# MVT: Multi-Vision Transformer for Event-Based Small Target Detection

**Shilong Jing** [1,2] , **Hengyi Lv** [1,*] , **Yuchen Zhao** [1] , **Hailong Liu** [1] **and Ming Sun** [1]

1 Changchun Institute of Optics, Fine Mechanics and Physics, Chinese Academy of Sciences, Changchun 130033, China; jingshilong22@mails.ucas.ac.cn (S.J.); zhaoyuchen@ciomp.ac.cn (Y.Z.); liuhailong@ciomp.ac.cn (H.L.); sunming@ciomp.ac.cn (M.S.);

2 University of Chinese Academy of Sciences, Beijing 100049, China

* Correspondence: lvhengyi@ciomp.ac.cn

**Abstract:** Object detection in remote sensing plays a crucial role in various ground identification tasks. However, due to the limited feature information contained within small targets, which are more susceptible to being buried by complex backgrounds, especially in extreme environments (e.g., low-light, motion-blur scenes). Meanwhile, event cameras offer a unique paradigm with high temporal resolution and wide dynamic range for object detection. These advantages enable event cameras without being limited by the intensity of light, to perform better in challenging conditions compared to traditional cameras. In this work, we introduce the Multi-Vision Transformer (MVT), which comprises three efficiently designed components: the downsampling module, the Channel Spatial Attention (CSA) module, and the Global Spatial Attention (GSA) module. This architecture simultaneously considers short-term and long-term dependencies in semantic information, resulting in improved performance for small object detection. Additionally, we propose Cross Deformable Attention (CDA), which progressively fuses high-level and low-level features instead of considering all scales at each layer, thereby reducing the computational complexity of multi-scale features. Nevertheless, due to the scarcity of event camera remote sensing datasets, we provide the Event Object Detection (EOD) dataset, which is the first dataset that includes various extreme scenarios specifically introduced for remote sensing using event cameras. Moreover, we conducted experiments on the EOD dataset and two typical unmanned aerial vehicle remote sensing datasets (VisDrone2019 and UAVDT Dataset). The comprehensive results demonstrate that the proposed MVT-Net achieves a promising and competitive performance.

**Keywords:** event cameras; multi-scale fusion; remote sensing; small target detection

## 1. Introduction

The event camera is a novel vision sensor inspired by biology, also known as Dynamic Vision Sensor (DVS) [1] or Dynamic and Active-Pixel Vision Sensor (DAVIS) [2]. Compared to conventional cameras that capture images at a fixed frame rate, event cameras independently measure and output the logarithmic intensity changes of each pixel instead of capturing images. When it comes to capturing fast-moving objects, traditional cameras require a significant cost to achieve satisfactory performance. In contrast, event cameras can effectively circumvent the limitations, providing asynchronous information with sub-millisecond latency. As a result, event cameras possess characteristics such as low latency, low power consumption, high dynamic range, and high temporal resolution. In addition, due to the fact that event cameras only capture changes in light intensity at different pixel locations, they can capture objects even in low-light conditions or extremely bright lighting. Thanks to these advantages, event cameras have demonstrated significant applications in both the military and civilian sectors. Figure 1 illustrates the theory of event generation in DVS.

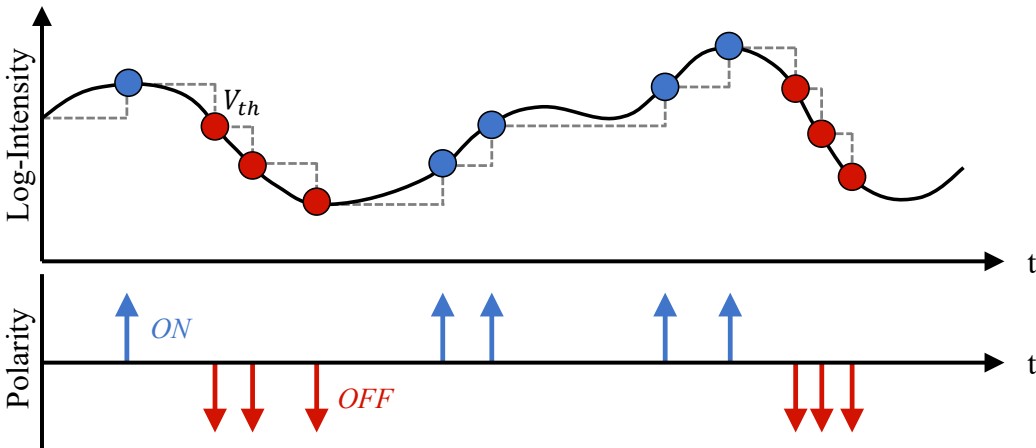

**Figure 1.** The process of DVS generates events. Each pixel serves as an independent detection unit for changes in brightness. An event is generated when the logarithmic intensity change at a pixel exceeds a specified threshold $V_th$. The continuous generation of events forms an event stream, which consists of two types of polarity: when the light intensity changes from strong to weak and reaches the threshold, DVS outputs a negative event (red arrow); when the light intensity changes from weak to strong and reaches the threshold, DVS outputs a positive event (blue arrow).

Utilizing drones equipped with event cameras for object detection or tracking is an innovative approach that holds great potential for a wide range of applications including satellite imaging, transportation, and early warning systems. However, due to the scarcity of remote sensing datasets based on the event cameras, we present the first event-based remote sensing dataset named Event-based Object Detection Dataset (EOD Dataset), which utilizes a DAVIS346 event camera mounted on an unmanned aerial vehicle (UAV) to capture various scenes. Furthermore, in practical processing, a high flying altitude results in ground targets occupying only a small portion of the image output, which poses challenges for object detection. Recently, advanced approaches for enhancing the detection performance of small targets often apply Feature Pyramid Networks (FPN) to concatenate multi-scale features. However, these methods have significant limitations as they are unable to differentiate between distinct feature layers. So how can we address this problem? Deformable DETR [3] provides an answer by introducing Scale-Level Embedding to differentiate the positional encoding of different features at the same location. Therefore, we draw inspiration from this embedding operation to concatenate multi-scale features, with the aim of enhancing the detection performance of small targets. Moreover, solely considering multi-scale features undoubtedly incurs significant computational and memory overhead, making convergence more challenging. For instance, in the Transformer Encoder of Deformable DETR, the model needs to extract features for all scales, even though deformable attention is used to reduce computational complexity, which still remains redundant.

In this work, we propose Cross-Deformable-Attention (CDA) to further enhance the performance of the model while significantly reducing its computational complexity. Specifically, by applying CDA between low-level and high-level features, we continuously propagate the fused information from lower layers to higher layers. In addition to reducing computational complexity, CDA can also reduce model training time and improve inference speed. What is more, we propose an efficient feature extraction model called Multi-Vision Transformer (MVT), which consists of three modules: Downsampling Module, Channel Spatial Attention Module (CSA), and Global Spatial Attention Module (GSA). Firstly, the downsampling module employs a simple overlapped convolution for scale reduction, resulting in better performance compared to non-overlapped convolution and patch merging operations. Then, we apply CSA for attention querying between spatial and channel dimensions. Compared to the original SE Block, CSA applies adaptive max pooling operations to preserve more high-frequency information. Finally, we employ GSA

including Window-Attention and Grid-Attention for local and global search. Compared to Swin-Attention, which requires more computational resources and complex offset vectors, Grid-Attention and Window-Attention are similar but only require local grid attention to extend them to the entire domain, achieving higher performance and fewer parameters. Additionally, we also provide three model variants (MVT-B, MVT-S, MVT-T) by setting different embedding dimensions and output scales. Employing MVT-B trained for 36 epochs, we achieve 28.7% mAP@0.5:0.95, outperforming all current state-of-the-art methods on the EOD dataset. With the application of multiple efficient attention modules that consider multi-scale features, the detection performance is improved especially for small objects, achieving 16.6% $AP_S$. While due to the scarcity of remote sensing datasets based on event cameras, we select the VisDrone2019 dataset [4] and UAVDT dataset [5], which are similar to our own dataset and consist of images captured by drones equipped with cameras. In this case, we employ MVT-B, which is trained for 36 epochs and achieve 31.7% mAP@0.5:0.95 and 24.3% $AP_S$ on the VisDrone2019 Dataset, as well as 28.2% mAP@0.5:0.95 and 23.7% $AP_S$ on the UAVDT Dataset.

Our contributions can be summarized as follows:

1. The first remote sensing dataset based on event cameras has been proposed, called the Event Object Detection Dataset (EOD Dataset), which consists of over 5000 event streams and includes six categories of objects like car, bus, pedestrian, two-wheel, boat, and ship.
2. We propose a novel multi-scale extraction network named Multi-Vision Transformer (MVT), which consists of three efficient modules proposed by us. The downsampling module, the Channel Spatial Attention (CSA) module, and the Global Spatial Attention (GSA) module. Overall, The MVT incorporates efficient modules, achieving a substantial reduction in computational complexity with high performance.
3. Considering that extracting information at all scales consumes massive computing resources, we propose a novel cross-scale attention mechanism that progressively fuses high-level features with low-level features, enabling the incorporation of low-level information. The Cross-Deformable-Attention (CDA) reduces the computational complexity of the Transformer Encoder and entire network by approximately 82% and 45% while preserving the original performance.
4. As a multi-scale object detection network, MVT achieves state-of-the-art performance trained from scratch without fine-tuning, which trained for 36 epochs, achieving 28.7% mAP@0.5:0.95 and 16.6% $AP_S$ on the EOD Dataset, 31.7% mAP@0.5:0.95 and 24.3% $AP_S$ on the VisDrone2019 Dataset, 28.2% mAP@0.5:0.95 and 23.7% $AP_S$ on the UAVDT Dataset.

## 2. Related Work

### 2.1. Multi-Scale Feature Learning

Convolutional neural networks extract features of objects through hierarchical abstractions, and an important concept in this process is the receptive field. Higher-level feature maps have larger receptive fields, which make them strong in representing semantic information, while they have lower spatial resolution and lack detailed spatial geometric features. On the other hand, lower-level feature maps have smaller receptive fields, which makes them strong in representing geometric details with higher resolution, but they exhibit weaker semantic information representation. For remote sensing object detection, the accuracy of small target recognition greatly affects the performance of the network. Therefore, multi-scale feature representation is a commonly used approach in small target detection [6,7].

The concept of the Feature Pyramid Networks (FPN) [8] is initially introduced for multi-scale object detection. However, the computation-intensive nature of the FPN significantly influences the detection speed. For this reason, various improvement methods have been developed. Centralized Feature Pyramid (CFP) [9] focuses on optimizing the representation of features within the same level, particularly in the corners of the im-

age. Path Aggregation Network (PANet) [10] extends the FPN with a bottom-up path to capture deeper-level features using shallow-level features. Additionally, the U-Net, originally designed for segmentation tasks, has also demonstrated outstanding performance in object detection [11–13].

In addition, there are methods that specifically utilize low-scale features for small target detection. Unlike approaches that recover high-resolution representation from low-resolution ones, the High-Resolution Network (HRNet) [6] maintains high-resolution representation during forward propagation. Lite-High-Resolution Network (Lite-HRNet) [14] can rapidly estimate feature points, thereby reducing the computational complexity of the model. Feature-Selection High-Resolution network (FSHRNet) [15] adopts HRNet as the backbone and introduces a Feature Selection Convolution (FSConv) layer to fuse multi-resolution features, enabling adaptive feature selection based on object characteristics. The Improved U-Net (IU-Net) [16] enhances the HRNetv2 [17] by incorporating the csAG module, composed of spatial attention and channel attention, to improve model performance. However, solely relying on low-scale features often leads to inferior performance, and the FPN operation fails to distinguish between different feature levels.

Scale-Level Embedding [3] was proposed for multi-scale fusion, which has the significant advantage of encoding different feature levels to enable the model to differentiate the same position information across different feature levels, and is widely applied in various types of models.

*2.2. Attention Mechanism*

The Attention Mechanism (AM) originated from studies on human vision. Due to the limitations in information processing, humans selectively focus on important information while disregarding less significant details [18]. In deep learning, AM is employed to mimic the human cognitive system by adding weights to different regions of feature maps, ensuring a prioritized processing order for neural networks [19,20]. Currently, AM can be broadly categorized into two branches: (1) applying pooling operations to extract salient information in channel or spatial dimensions [21]; (2) employing self-attention mechanisms to model global information and capture long-range dependencies [19].

There are several representative approaches in the first branch. Squeeze-and-Excitation Networks (SENet) [22] operate in the channel dimension, applying global pooling and fully connected layers to downsample feature maps to a single point and employ a multilayer perceptron (MLP) to generate weights for different regions. Then, the Hadamard product is computed between the weights applied sigmoid activation function and the original input to obtain channel-weighted feature maps, establishing relationships between channels. Efficient Channel Attention Networks (ECA-Net) [20] is an improved version of SENet that uses 1D convolution instead of fully connected layers to achieve channel-wise information interaction, which avoids the degradation of a part of feature representations during the scale variation process. The Convolutional Block Attention Module (CBAM) [23] further introduces the Spatial Attention Module (SAM), which calculates weights for both the channel and spatial domains, selectively assigning importance to different features. SAM first generates distinct global information feature maps through pooling operations. Subsequently, the Hadamard product is computed between the result applied sigmoid and the original input to enhance the target region. Due to the lightweight and plug-and-play advantages, these methods have been widely applied. However, their drawback lies in the loss of features for small objects due to their limitations in long-range regions.

Transformer [19] is the representative approach in the second branch, capable of effectively extracting features from long-range regions. The Vision Transformer (VIT) [24] provides a novel approach to extract features by treating images as tokens, similar to sentences, to capture global information. Due to its simplicity and strong scalability, sparking subsequent research. However, VIT still faces the challenge of high computational complexity with excessively long tokens. Therefore, VIT only extracts features from images with an input resolution of 224. To solve these problems, Swin Transformer [25]

introduces a window shift strategy to overcome the limitation of input resolution and utilizes a window sliding mechanism with convolutional operations to enable interaction between different windows, thus achieving global attention. Despite achieving remarkable results in various tasks, the Swin Transformer still faces the redundancy of using offset vectors. Furthermore, Multi-Axis Vision Transformer (MAXVIT) [26] proposes Multi-axis Self-Attention (MaxSA), which decomposes the conventional self-attention mechanism into two sparse forms: Window-Attention and Grid-Attention. This approach reduces the quadratic complexity of traditional computation methods to linear complexity. Importantly, it discards redundant window offset operations and instead employs a simpler form of window attention and grid attention to consider both local and global information. Additionally, Deformable DETR [3] introduces Deformable-Attention, which can be summarized as each feature pixel does not need to interact with all other feature pixels for computation. Instead, it only needs to interact with a subset of other pixels obtained through sampling. This mechanism significantly accelerates model convergence while reducing computational complexity. The aforementioned studies discuss the capability of Transformer Attention to model global information for accurate target localization. While these methods have made improvements in terms of computational resources, they still encounter challenges regarding the excessive computational complexity caused by remote sensing images. Therefore, we propose a novel Cross-Deformable-Attention (CDA) structure to achieve a balance between performance and computational cost.

*2.3. Remote Sensing Images Object Detection*

Currently, the mainstream frameworks for event camera object detection include CNN-based [27–29] and Transformer-based [3,30,31] approaches. Specifically, the event streams are encoded into spatiotemporal tensors, which are then fed into deep neural networks for some complex downstream tasks. While this process is similar to conventional image detection frameworks, the representation of the event tensor is significantly different from the image. Therefore, the performance of the network is directly influenced by the extracted information. Meanwhile, RNN-based [32] models have also shown great potential in event camera detection.

The small targets in remote sensing are often susceptible to interference from complex backgrounds. There are several studies have shown that enhancing multi-scale features can significantly improve small target detection. Compared to R-CNN [33] and Faster R-CNN [34], which generate redundant bounding boxes during small object detection, Events-SSD [35] introduces Single-Shot MultiBox Detector to improve detection efficiency. However, due to its relatively weak representation capability in shallow feature maps, it is not robust for small targets. Events-YOLO [36] improves upon Events-SSD by introducing a multi-scale detection mechanism that combines visible frames to supplement event representations with finer details, enabling the detection of objects at different scales. RVT [32] introduces a novel recurrent neural network that incorporates the temporal dimension of event tensors, achieving excellent performance on ground vehicle datasets. EMS-YOLO [37] directly trains a deep Spiking Neural Network, aiming for better applicability to neuro-computing hardware by binary data communication. While RVT and EMS-YOLO both take into account the temporal sequence of the event stream, they are frameworks in the field of ground object detection, utilizing FPN for multi-scale fusion rather than Scale-Level Embedding that can differentiate information of different scales.

In general, event cameras have seen emerging developments in ground-based detection, while research in the field of remote sensing remains notably scarce. Moreover, due to the extreme challenges (e.g., smaller targets, more severe motion blur, more complex backgrounds) associated with event camera remote sensing detection, designing an efficient backbone becomes particularly crucial. However, existing research lacks the capability of global modeling, and is unable to extract long-dependence information, especially in high-resolution remote sensing images. In this work, our objective is to propose a novel end-to-end object detection framework that better inherits the advantages of multi-scale fea-

tures and attention mechanisms to address small target detection in complex backgrounds of remote sensing.

## 3. Method

### 3.1. Overall Architecture

The proposed MVT Network is illustrated in Figure 2, which is composed of four main components, namely Data Processing, MVT Backbone, Feature Fusion Module, and Prediction Head.

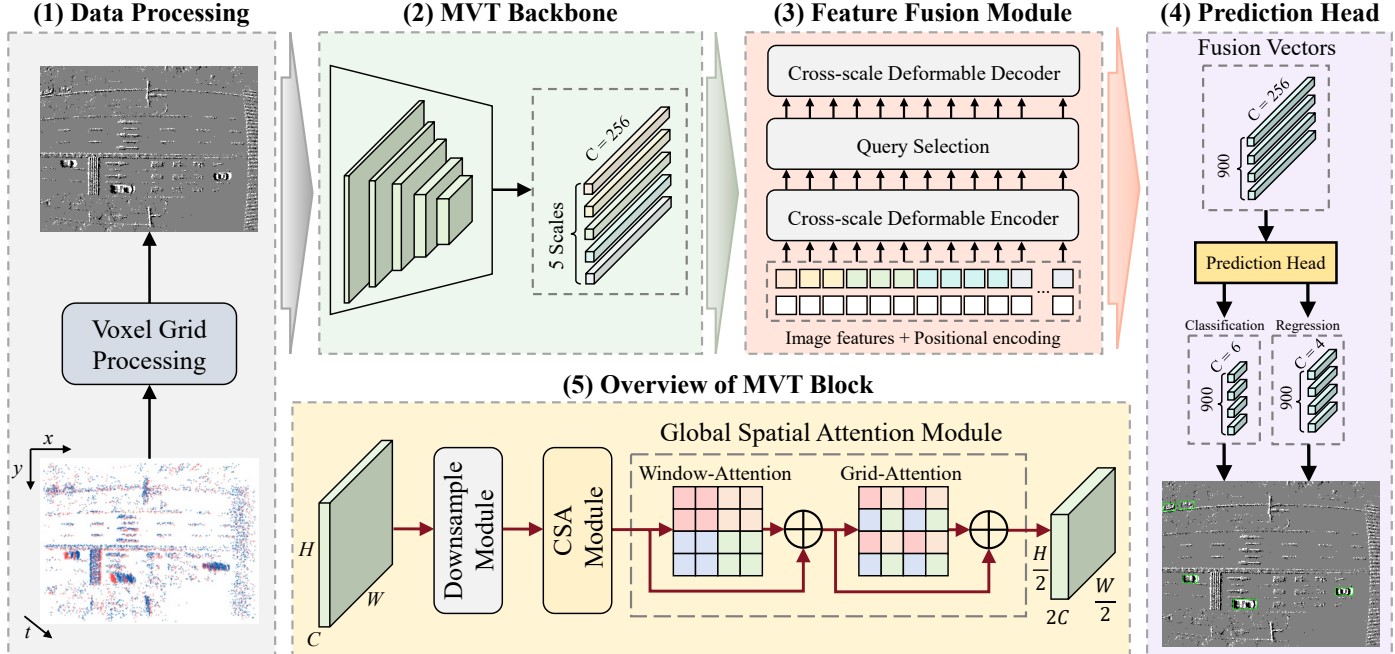

**Figure 2.** Overview of the MVT framework, which contains five main components, including: (1) the data preprocessing method of converting event streams into dense tensors; (2) the proposed MVT Backbone used to extract multi-scale features; (3) the designed feature fusion module for encoding and aggregating features at different scales; (4) the detection head that applies bipartite matching strategy; (5) Each MVT Block, composed of three designed components.

The original chaotic event sequence cannot be directly used as an input tensor for deep neural networks. Therefore, we encode the event stream in the form of voxel grid representation [38], which has a channel regarding the temporal dimension generated by a time partitioning function, and described in detail in Section 3.2. In this work, we do not consider the correlation of the temporal order. Thus, the processed event tensor has a shape of $X = \mathbb{R}^{H \times W \times 1}$. Different scale features of the event tensor are extracted by the backbone, which utilizes CSA to attend to short-range dependencies and GSA to attend to long-range dependencies, which is specifically described in Section 3.3. Subsequently, the multi-scale features with rich semantic information are fed into the Transformer Encoder, where CDA is applied to fuse tokens at different levels, which is described in detail in Section 3.4. Finally, regression calculations are performed on the 900 vectors generated by the Feature Fusion Module to obtain the detection results.

### 3.2. Event Representation

The event camera captures the brightness changes of individual pixels, generating an asynchronous event stream. An event with polarity is generated at time $t$ when the logarithmic change of light intensity $I^t(u)$ exceeds the threshold $V_{th}$ within a small time interval $\Delta t$, which satisfies

$$p[I^t(u) - I^{t-\Delta t}(u)] \geqslant V_{th} \tag{1}$$

where $p \in \{0, 1\}$ is the event polarity, $V_{th}$ is the threshold. The event camera will generate an ordered set of events $\varepsilon = \{e_k\}^{E_x, E_y, E_p, E_t} \in \mathbb{R}^4$ according to Equation (1). Afterwards, the polarity of each pixel in the same time window is aggregated by performing bilinear voting, which requires the standardization of event timestamps as

$$E_{t\_norm} = T \frac{E_t - E_t(0)}{E_t(N) - E_t(0)} \tag{2}$$

where $T$ represents the number of non-overlapping time windows, which we set to 1 in this work. $E_t(N)$ indicates the timestamp corresponding to the last event. Equation (2) demonstrates the normalization of temporal dimensions for the event stream. In addition, the two encoded polarities are represented as

$$E_{p\_left} = E_p(E_{t\_norm} - [E_{t\_norm}]) \tag{3}$$

$$E_{p\_right} = E_p(1.0 - (E_{t\_norm} - [E_{t\_norm}])) \tag{4}$$

where $[\cdot]$ represents the floor function. Equations (3) and (4), respectively, denote the product of the time distance from the current event to the start and end points of the time window and the polarity. Finally, by accumulating the encoded polarities at the corresponding pixel position $(E_x, E_y)$, we obtain the event tensor in the form of voxel grid representation. The Algorithm 1 for event representation is as follows:

---

**Algorithm 1** Voxel grid encoding from event stream

---

**Input:** Event stream containing N number of events $\varepsilon = \{e_k\}^{E_x, E_y, E_p, E_t} \in \mathbb{R}^4$.
**Output:** Voxel grid tensor $X = \mathbb{R}^{H \times W}$.
1: $X = \mathbb{R}^{H \times W}$; // Create a tensor with all values set to 0;
2: Compute the normalized event stream time $E_{t\_norm}$ according to Equation (2);
3: $TI = [E_{t\_norm}]$; // Perform time windowing based on the setting values;
4: Compute the encoded polarity fused event time $E_{p\_left}$ and $E_{p\_right}$ according to Equations (3) and (4);
5: **if** $(TI < T)$ **then**
6:     **for** $(i = 0, i < len(T), i++)$ **do**
7:         $X(E_x[i], E_y[i]) += E_{p\_left}[i]$; // Accumulate the left polarity at the corresponding pixel positions where events occur.
8:     **end for**
9: **end if**
10: **if** $(TI + 1 < T)$ **then**
11:     **for** $(i = 0, i < len(T), i++)$ **do**
12:         $X(E_x[i], E_y[i]) += E_{p\_right}[i]$; // Accumulate the right polarity at the corresponding pixel positions where events occur.
13:     **end for**
14: **end if**
15: **return** $X$

---

### 3.3. Multi-Vision Transformer (MVT)

The MVT Backbone consists of four layers, with each layer stacked with a varying number of MVT Blocks to extract features at different scales, which is specifically demonstrated in illustration (5) of Figure 2. The MVT Block consists of three components: the downsampling module applying overlapping convolutions, the CSA module utilizing spatial and channel attention to consider short-term attention, and the GSA module employing Window-Attention and Grid-Attention to consider long-term attention.

#### 3.3.1. Downsample Module

We design an extremely simple downsampling module, which consists of an overlapping convolution. Specifically, in the first layer, we use a $7 \times 7$ convolution kernel

with a stride of 4 to achieve fourfold downsampling, while the remaining layers apply a $3 \times 3$ convolution kernel with a stride of 2 for two-fold downsampling. Furthermore, we demonstrate that the overlapping convolution outperforms non-overlapping convolutions and patch merging operations in Section 4.3.

### 3.3.2. Channel Spatial Attention Module (CSA)

In this section, we introduce CSA for extracting short-term dependency attention, which assigns more weight to focal channels and spatial locations in the feature map, thereby enhancing the capability of feature representation, as illustrated in Figure 3.

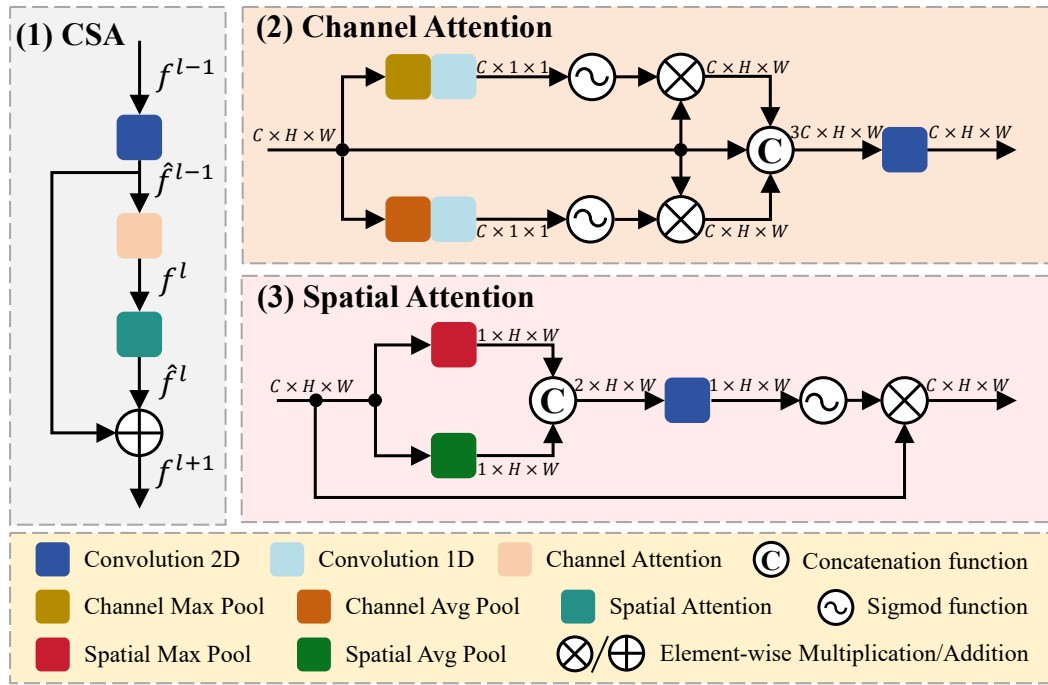

**Figure 3.** Architecture of CSA module, which consists of channel attention and spatial attention module to extract short-term dependent attention.

The input feature map $f^{l-1}$ is fed into CSA for feature extraction. Firstly, $f^{l-1}$ undergoes a 2D convolution (*Conv*2*D*) operation with a $1 \times 1$ kernel, resulting in $\widehat{f}^{l-1}$, which has the same dimensions as $f^{l-1}$. Then, $\widehat{f}^{l-1}$ is separately fed into channel attention ($\mathcal{L}^C_{Attn}$) and spatial attention ($\mathcal{L}^S_{Attn}$) modules, producing the intermediate feature map $\widehat{f}^l$, which is added to $\widehat{f}^{l-1}$ to obtain the output feature map $f^{l+1}$. The entire CSA computation process can be represented by Equation (5).

$$\begin{aligned} \widehat{f}^{l-1} &= Conv2D(f^{l-1}) \\ \widehat{f}^l &= \mathcal{L}^S_{Attn}(\mathcal{L}^C_{Attn}(\widehat{f}^{l-1})) \\ f^{l+1} &= \widehat{f}^{l-1} \oplus \widehat{f}^l \end{aligned} \tag{5}$$

The main components of CSA can be divided into Channel Attention and Spatial Attention. Within the Channel Attention module, there are three branches: in the first branch, the input ($F$) is fed into Channel Max Pooling ($\mathcal{P}^C_{Max}$) and a 1D convolution (*Conv*1*D*), resulting in a tensor ($F_{max}$) with $C \times 1 \times 1$ dimensions, then, applying a sigmoid function ($\sigma$) to obtain attention weights, which are multiplied with the shortcut layer to produce a feature map ($\widehat{F}_{max}$) with $C \times H \times W$ dimensions; the second branch utilizes a residual connection to preserve the original features, enhancing the representational and generalization capabilities of the network; the third branch differs from the first branch only in applying Channel Average Pooling ($\mathcal{P}^C_{Avg}$) to extract features ($\widehat{F}_{avg}$). In addition, a concate-

nation function (*Concat*) is employed to transform the three feature maps with $C \times H \times W$ dimensions into a single feature map with $3C \times H \times W$ dimensions. Finally, utilizing a 2D convolution to map the channels back to $C \times H \times W$ and obtain the feature map ($F_{Attn}^C$). The entire Channel Attention computation process can be represented by Equation (6).

$$
\begin{aligned}
F_{max} &= Conv1D(\mathcal{P}_{Max}^C(F)) \\
\widehat{F}_{max} &= \sigma(F_{max}) \otimes F \\
F_{avg} &= Conv1D(\mathcal{P}_{Avg}^C(F)) \\
\widehat{F}_{avg} &= \sigma(F_{avg}) \otimes F \\
F_{Attn}^C &= Conv2D(Concat[F, \widehat{F}_{max}, \widehat{F}_{avg}])
\end{aligned}
\tag{6}
$$

Within the Spatial Attention module, there are two branches: in the first branch, the input ($F$) is fed into both Spatial Max Pooling ($\mathcal{P}_{Max}^S$) and Spatial Average Pooling ($\mathcal{P}_{Avg}^S$) to obtain features ($\widehat{F}_{max}$) and ($\widehat{F}_{avg}$), which are then concatenated to form a tensor ($\widehat{F}$) with $2 \times H \times W$ dimensions. Next, the feature map ($\widehat{F}$) undergoes a 2D convolution (*Conv2D*) followed by a sigmoid function ($\sigma$), resulting in spatial attention weights ($\widehat{F}_{Attn}^S$), which are multiplied element-wise with the original input ($F$) to achieve the final feature map ($F_{Attn}^S$) in the second branch. The entire Spatial Attention computation process can be represented by Equation (7).

$$
\begin{aligned}
\widehat{F} &= Concat[\mathcal{P}_{Max}^S(F), \mathcal{P}_{Avg}^S(F)] \\
\widehat{F}_{Attn}^S &= \sigma(Conv2D(\widehat{F})) \\
F_{Attn}^S &= \widehat{F}_{Attn}^S \otimes F
\end{aligned}
\tag{7}
$$

The CSA module improves the feature extraction performance for short-range regions by incorporating attention mechanisms for both channels and spatial dimensions. However, convolutional attention modules suffer from a loss of features for small objects due to their limitations in long-range regions. Therefore, we propose GSA, considering global attention to enhance the detection performance of small targets.

### 3.3.3. Global Spatial Attention Module (GSA)

In this section, we introduce GSA for extracting long-term dependency attention, which is able to obtain global information and long-distance connections in one single operation, as illustrated in Figure 4.

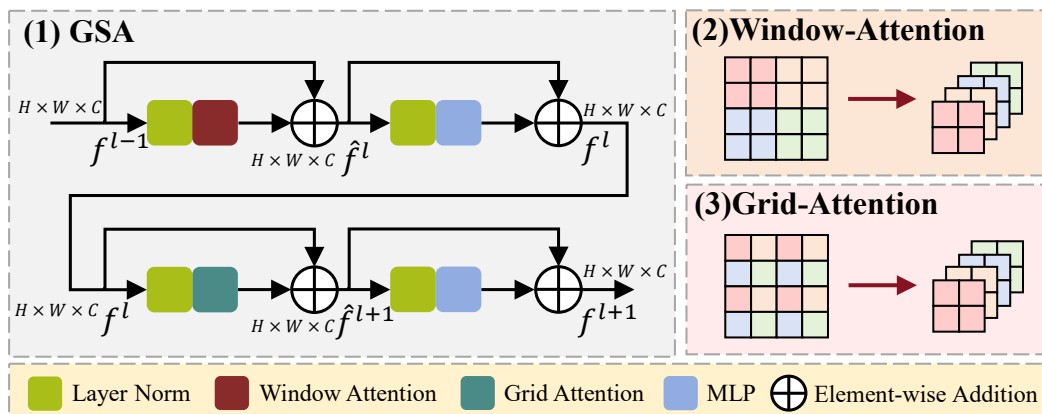

**Figure 4.** Architecture of GSA module, which consists of window attention and grid attention to extract long-term-dependent attention.

We propose an efficient modeling solution with two window configurations: Window-based Multi-head Self-Attention (W-MSA) and Grid-based Multi-head Self-Attention (G-MSA). Firstly, the input ($f^{l-1}$) is normalized and fed into the W-MSA to obtain the local

attention feature map, which is then added to the original input ($f^{l-1}$) through a residual path, resulting in the hidden feature map ($\hat{f}^l$). Subsequently, ($\hat{f}^l$) is separately processed through Layer Normalization (LN) + Multilayer Perceptron (MLP) and the shortcut path to obtain the feature map ($f^l$). In addition, ($f^l$) undergoes LN and G-MSA to obtain the global attention feature map ($\hat{f}^{l+1}$), which is further processed through LN and MLP to obtain the global spatial feature map ($f^{l+1}$). The entire Global Spatial Attention computation process can be represented by Equation (8).

$$
\begin{aligned}
\hat{f}^l &= \text{W-MSA}(LN(f^{l-1})) + f^{l-1} \\
f^l &= MLP(LN(\hat{f}^l)) + \hat{f}^l \\
\hat{f}^{l+1} &= \text{G-MSA}(LN(f^l)) + f^l \\
f^{l+1} &= MLP(LN(\hat{f}^{l+1})) + \hat{f}^{l+1}
\end{aligned}
\tag{8}
$$

### 3.4. Cross Deformable Attention (CDA)

The framework of the Cross-scale Deformable Attention (CDA) is shown in Figure 5. Different from the repeated iterative feature extraction operation of multi-scale cross fusion, we propose CDA to achieve layer-by-layer feature fusion to better fuse feature maps of different scales and reduce computational complexity. Accordingly, enhances the representation of high-level features with both high-level semantics and high-resolution details.

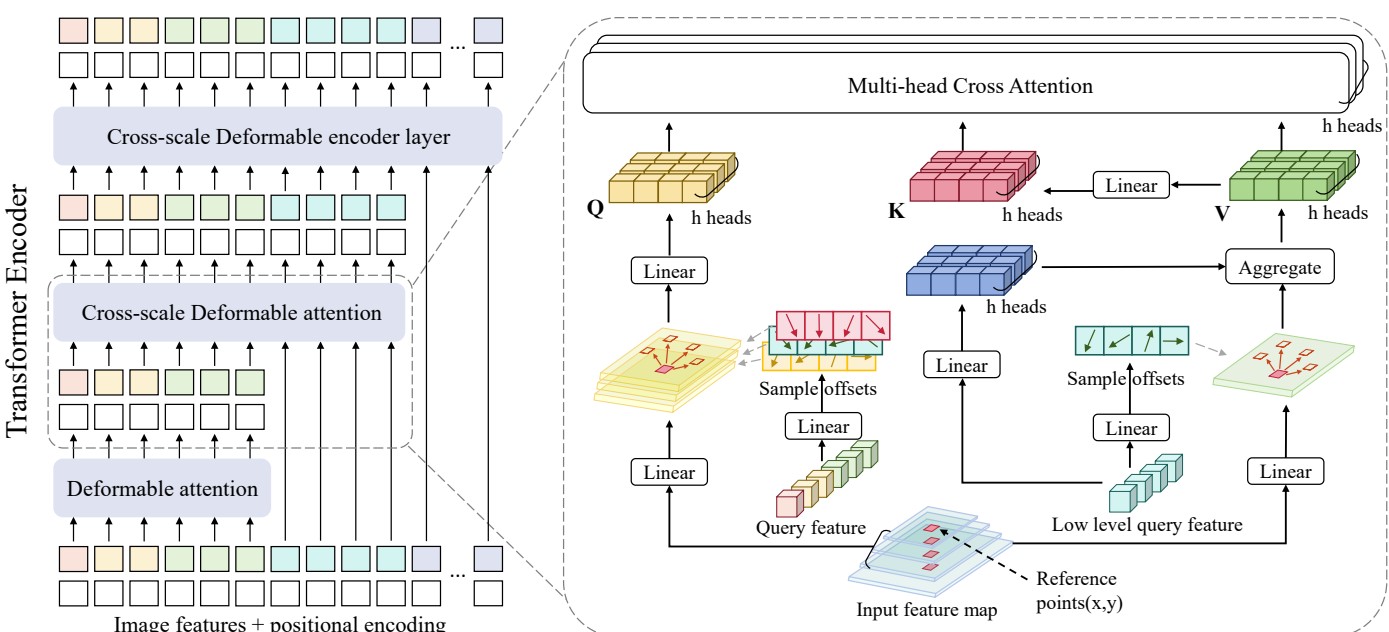

**Figure 5.** Overview of the Cross-scale Deformable Encoder layer. The three high-level features are used as the basic tokens to fuse low-level features layer by layer using Cross-scale Deformable Attention, finally building the architecture of the transformer encoder.

The encoder layer contains deformable self-attention and cross-scale attention. Considering that the feature map size of the high level is much smaller than the low level. Thus only the middle and final encoder layers are needed for cross-scale attention to the low and high scale instead of extracting all tokens, as shown in Figure 5. In this module, high-level features $\mathbf{F_H} \in \mathbb{R}^{N_H \times d_{model}}$ will serve as queries to extract features from the low-level features $\mathbf{F_L} \in \mathbb{R}^{N_L \times d_{model}}$, each query feature will be split into $M$ heads, and each head will sample $K$ points from each of the $L$ feature scales as query $\mathbf{Q}$. Therefore, the total number of queries sampled for a query feature is $N_p = 2 \times M \times L \times K$, $\Delta p$ is sampling offsets, and their

corresponding attention weights are directly predicted from query features using two linear projections denoted as $W_p \in \mathbb{R}^{d_{model} \times N_p}$ and $W_A \in \mathbb{R}^{d_{model} \times d_{model}}$. Formally, we have

$$\mathbf{Q} = \sum_{m=1}^{M} W_m [\sum_{l=1}^{L} \sum_{k=1}^{K} W'_m S \left( x^l, \phi(p^l) + \Delta p_{mlk} \right)] \tag{9}$$

$$\mathbf{K} = \sum_{m=1}^{M} W_m [\sum_{k=1}^{K} W_A \cdot W'_m S(x, p + \Delta p_{mk})] \tag{10}$$

where $m$ is the attention head, $p$ are the reference points of the query features, $x$ indexes the different scale feature, $W_m \in \mathbb{R}^{d_{model} \times N_m}$ and $W'_m \in \mathbb{R}^{N_m \times d_{model}}$ are of learnable weights ($N_m = d_{model} / M$ by default). With the sampled offsets ($\Delta p = \mathbf{F} W_p$), bilinear interpolation is applied in computing the features with the function $S(x, p + \Delta p)$ in the sampled locations ($p + \Delta p$) of the corresponding feature $x$. As all the high-level features will sample locations to query the key consisting of low-level features, the original model can quickly learn which sampled location given the queries is important. Finally, we can obtain the value ($\mathbf{V} = \mathbf{K} W_V$) with a parameter matrices $W_V \in \mathbb{R}^{d_{model} \times d_{model}}$, and the cross-scale deformable attention can be formulated as

$$CDA(\mathbf{Q}, \mathbf{K}, \mathbf{V}) = Cat(\mathbf{F_L}, Softmax(\frac{\mathbf{Q} \mathbf{K}^T}{\sqrt{d_K}}) \mathbf{V}) \tag{11}$$

In words, the cat function is to concatenate low-level features and other multi-scale features, $d_k$ is the key dimension of a head. Equation (11) indicates more reliable attention weights predicted by stacking CDA when updating layer-by-layer features from different scales.

## 4. Experiments

In this section, we test the proposed method on the EOD, VisDrone [4], and UAVDT [5] datasets, and the mean average precision (mAP) [39] is the main metric that we consider. In addition, we perform ablation experiments to verify the effectiveness of each module. Finally, the experimental results demonstrate the superiority of the proposed method.

### 4.1. Datasets

#### 4.1.1. EOD Dataset

The EOD dataset consists of 5317 event streams captured in various scenes, where each event stream is a collection of events within 33 ms. The dataset includes 3722 event streams for training, 530 event streams for validation, and 1065 event streams for testing, and contains six categories: car, bus, pedestrian, two-wheel, boat, and ship.

#### 4.1.2. VisDrone Dataset

The VisDrone-DET2019 dataset [4] consists of 8599 images, including 6471 images for training, and 1580 images for testing. The dataset contains ten categories: person, pedestrian, car, bus, truck, bicycle, tricycle, awning-tricycle, van, and motor.

#### 4.1.3. UAVDT Dataset

The UAVDT dataset [5] consists of 40,409 images, selected from 10 h long videos that cover various scene variations (e.g., weather, viewpoint, and illumination), including 23,829 images for training and 16,580 images for testing. The images in this dataset have a resolution of $540 \times 1024$ pixels and include three categories: car, bus, and truck.

### 4.2. Implementation Details

#### 4.2.1. Evaluation Metrics

We quantitatively evaluate the performance of our method through the mAP, which is used to comprehensively evaluate the precision and recall of a model across different categories, commonly used in object detection. Specifically, the mAP can be defined as the

area under the precision–recall (P-R) curve when plotted with the recall (R) on the horizontal axis and precision (P) on the vertical axis. mAP@0.5 refers to the IOU (Intersection of Union) is greater than 0.5. mAP@0.5:0.95 refers to the average of IOU values from 0.5 to 0.95 with an interval of 0.05. The P and R are defined as

$$
\begin{aligned}
Precision &= \frac{TP}{TP + FP} \\
Recall &= \frac{TP}{TP + FN}
\end{aligned}
\tag{12}
$$

where $TP$ (True Positive) indicates the number of positive samples correctly classified as positive by the model, $FP$ (False Positive) represents the number of negative samples incorrectly classified as positive by the model, and $FN$ (False Negative) represents the number of positive samples incorrectly classified as negative by the model. By calculating the area under the P-R curve, the mAP is defined as

$$
mAP = \int_0^1 P(R)dR
\tag{13}
$$

where $P(R)$ is a function of $P$ and $R$. In addition, we also evaluate the model size and computational complexity through Params and GFLOPs (giga floating point of operations).

### 4.2.2. Training Settings

In this work, we use a 6-layer Transformer encoder and a 6-layer Transformer decoder, utilize 8 as multi-heads and 4 as sampling offset, adopt 2048 as Transformer feed-forward, and 256 as the hidden feature dimension, apply $1 \times 10^{-4}$ as the initial learning rate and $1 \times 10^{-5}$ as the backbone learning rate. In addition, we use the AdamW optimizer with a weight decay of $1 \times 10^{-4}$ and train our model by using the PyTorch framework with 8 Nvidia GeForce RTX3090 GPUs on Ubuntu22.04 with batch size 32 for all datasets. Particularly, our model is trained from scratch without pre-training and fine-tuning.

### 4.2.3. Model Variants

By setting different dimensions and final output scales for each layer, we constructed three variants of MVT-B/S/T. Where MVT-B is the base form with five stages, MVT-S and MVT-T are small and tiny forms with four and three stages, respectively. After the backbone feature extraction, the feature maps are missing the final stage that is obtained by applying a convolutional block to the last-second feature map. Furthermore, the first layer utilizes a $7 \times 7$ kernel with a stride of 4 for overlapping convolution to reduce the input feature resolution and computational cost. The second to fourth layers adopt a $3 \times 3$ kernel with a stride of 2 for overlapping convolution to extract higher-level feature information. "✓" indicates that the stage serves as a multi-scale feature output. Table 1 shows the specific parameters for different variants.

**Table 1.** MVT parameters and variations. Except for the channel numbers at each stage, all model variants share the same parameter set.

| Stage | Size | Kernel | Stride | Channels | | | | | |
|---|---|---|---|---|---|---|---|---|---|
| | | | | MVT-B | | MVT-S | | MVT-T | |
| S1 | 1/4 | 7 | 4 | 96 | ✓ | 64 | | 32 | |
| S2 | 1/8 | 3 | 2 | 192 | ✓ | 128 | ✓ | 64 | |
| S3 | 1/16 | 3 | 2 | 384 | ✓ | 256 | ✓ | 128 | ✓ |
| S4 | 1/32 | 3 | 2 | 768 | ✓ | 512 | ✓ | 256 | ✓ |

We utilize three variants, MVT-B/S/T, for detection on the EOD dataset. Figure 6 presents the detection results in different scenarios. Since the event camera outputs asyn-

chronous data, and generates corresponding events even in low light and overexposure without limitation by the intensity of light.

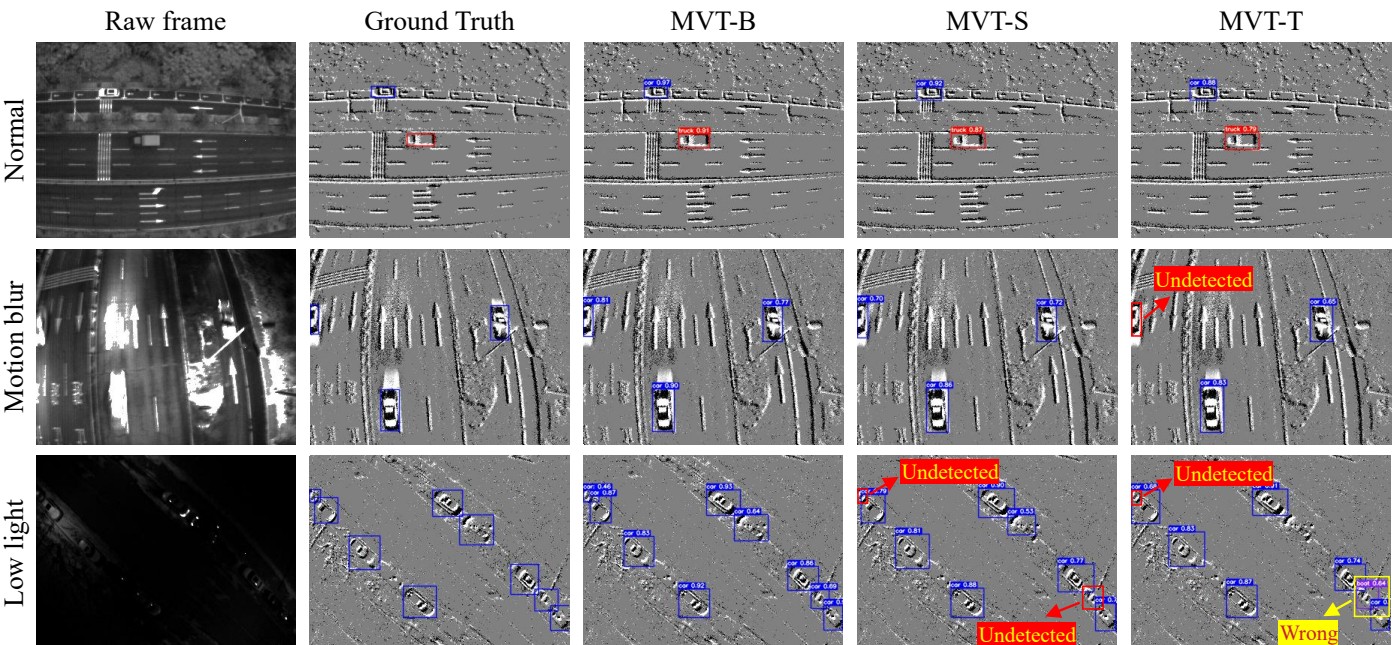

**Figure 6.** Prediction examples on the EOD dataset. The MVT-B/S/T variants are applied to detect in normal, motion blur, and low-light scenarios, respectively.

MVT-B outperforms the other variants due to its higher-resolution feature scale information, resulting in superior performance in detecting small objects. Specifically, in scenarios with motion blur and low light, MVT-S and MVT-T occasionally fail to detect small targets located in the top-left corner. However, despite the inherent advantages of event cameras over traditional cameras in terms of efficiency, they suffer from the loss of high-frequency information in the images, leading to the degradation of image details. Consequently, under low-light conditions, MVT-T erroneously misclassifies a car as a boat.

### 4.3. Ablation Experiments

In this section, we conducted ablation experiments on the EOD dataset to assess the contribution of each proposed module to the results. The contribution to the final results by evaluating the detection performance before and after applying each module. Table 2 represents the performance of ablation experiments.

**Table 2.** Ablation experiment on the EOD dataset. "✓" indicates that the module is used in the MVT network, while "-" indicates that it is not used, best results in **bold**, underlined denotes the second best performance, and the same colors indicate the same benchmarks except for CDA.

| Model | Structure | | | mAP @0.5:0.95 | mAP @0.5 | Entire GFLOPs | Encoder GFLOPs | Params |
|---|---|---|---|---|---|---|---|---|
| | CSA | GSA | CDA | | | | | |
| Baseline | - | - | - | 0.214 | 0.403 | 67.6 | 47.6 | 25.6 M |
| MVT-B | ✓ | - | - | 0.238 | 0.474 | 69.7 | 47.6 | 34.5 M |
| | - | ✓ | - | 0.265 | 0.527 | 84.7 | 47.6 | 97.3 M |
| | - | - | ✓ | 0.212 | 0.401 | **28.4** | **8.4** | 25.7 M |
| | ✓ | ✓ | - | **0.288** | **0.569** | 86.8 | 47.6 | 106.3 M |
| | ✓ | ✓ | ✓ | <u>0.287</u> | <u>0.565</u> | 47.6 | **8.4** | 106.4 M |

By progressively incorporating the designed modules, we achieve higher mAP. Compared to the baseline that only considers downsampling, the inclusion of CSA for extracting

channel and spatial information improves mAP@0.5:0.95 by 2.4%, the incorporation of GSA for extracting global spatial information enhances mAP@0.5:0.95 by 5.1%, the introduction of CDA reduces model computational complexity by approximately 58% in terms of GFLOPs while maintaining the original performance. Combining CSA and GSA results in a 7.4% increase in mAP@0.5:0.95. Finally, by considering CSA, GSA, and CDA, we achieve 28.7% mAP@0.5:0.95, reducing Entire GFLOPs and Encoder GFLOPs by approximately 45% and 82% compared to the model without CDA. Figure 7 shows the attention visualization both without and with CDA.

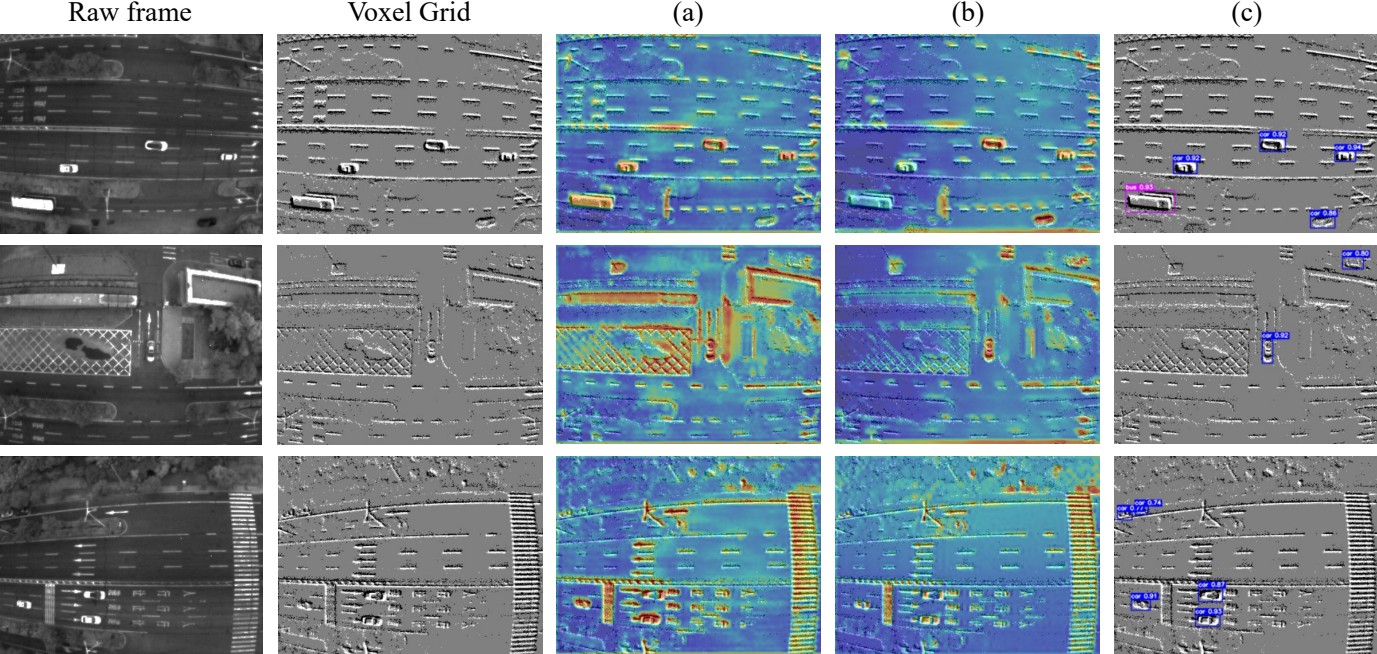

**Figure 7.** Visualization of attention maps. (**a**) Visualization of feature maps generated by the model without CDA. (**b**) Visualization of feature maps generated by the model with CDA. It can be observed that the attention applied by CDA is more focused on small targets. (**c**) Detection results applied CDA.

### 4.3.1. Ablation of Downsample Module

In the original Vision Transformer [24], downsampling operations primarily employ patch merging, which involves fewer parameters compared to pooling layers while fully preserving the input feature maps. Specifically, elements are divided and concatenated with two strides in both rows and columns, and a linear layer is utilized to scale the input feature maps from 4× to 2×. Thus, we have separately compared patch merging, overlapping, and non-overlapping convolutional downsampling blocks. Table 3 demonstrates that the use of convolutional downsampling outperforms patch merging.

**Table 3.** Ablation of the downsampling module. Best results in **bold**. The usage of Conv. overlapping outperforms other downsample approaches.

| Downsampling Type | mAP @0.5:0.95 | mAP @0.5 | mAP @0.75 | $mAP_S$ | $mAP_M$ | $mAP_L$ | Params |
|---|---|---|---|---|---|---|---|
| Patch Merging | 0.281 | 0.557 | 0.254 | 0.159 | 0.336 | 0.566 | 6.21 M |
| Conv. non-overlapping | 0.283 | 0.559 | 0.257 | 0.160 | 0.337 | 0.565 | **6.20** M |
| Conv. overlapping | **0.290** | **0.573** | **0.264** | **0.166** | **0.358** | **0.582** | 13.94 M |

### 4.3.2. Ablation of GSA Module

We apply Swin-Attention [25] and Grid-Attention [26] as global attention modules, respectively, to consider the all tokens. Table 4 shows that using Swin-Attention consumes more computational resources and has lower performance than Grid-Attention.

**Table 4.** Ablation of the global spatial attention module. Best results in **bold**. The usage of Grid-Attention outperforms Swin-Attention.

| Attention Type | mAP @0.5:0.95 | mAP @0.5 | mAP @0.75 | mAP$_S$ | mAP$_M$ | mAP$_L$ | Params |
|---|---|---|---|---|---|---|---|
| Swin-Attn | 0.280 | 0.558 | 0.251 | 0.162 | 0.347 | 0.545 | 99.5 M |
| Grid-Attn | **0.287** | **0.565** | **0.263** | **0.166** | **0.353** | **0.580** | **66.9** M |

4.3.3. Effect of Multi-Vision Transformer Network

The introduction of CSA and GSA aims to efficiently extract features at different scales while considering dependencies between short-range and long-range features. The second and third rows in Table 1 demonstrate the effects of incorporating CSA and GSA into the baseline respectively. CSA improves mAP@0.5:0.95 by 2.4% and mAP@0.5 by 7.1% with a slight increase in the number of GFLOPs and parameters. GSA improves mAP@0.5:0.95 by 5.1% and mAP@0.5 by 12.4% while at the cost of higher GFLOPs and parameters. Furthermore, the joint utilization of CSA and GSA results in an improvement of 7.4% mAP@0.5:0.95 and 16.6% mAP@0.5. By introducing CDA, the computational complexity of the model was reduced by approximately 58% and 45% compared to the baseline and the model that combines CSA and GSA. Figure 8 provides a visual comparison of the detection results obtained by MVT when using CSA alone, GSA alone, and both CSA and GSA.

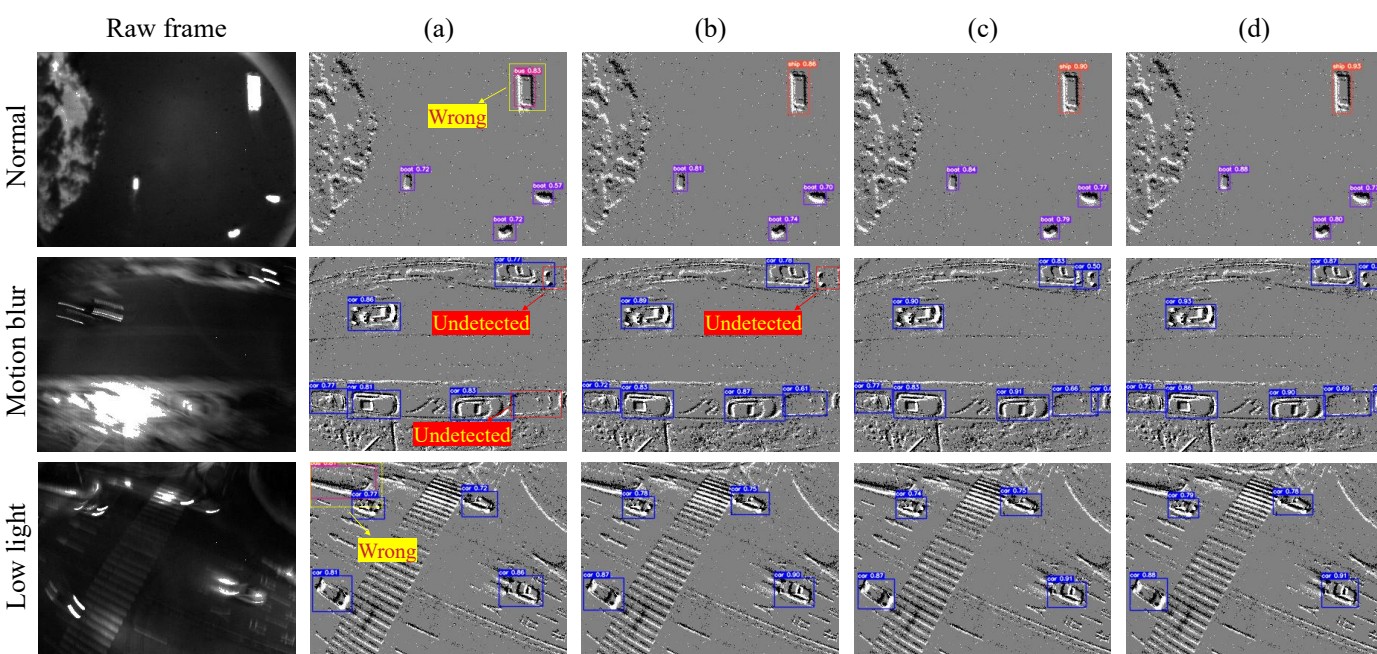

**Figure 8.** Comparison of the detection results before and after using CSA alone, GSA alone, and both CSA and GSA in the MVT network. (**a**) Baseline. (**b**) Baseline + CSA. (**c**) Baseline + GSA. (**d**) Baseline + CSA + GSA.

There are significant visual improvements as the baseline progressively incorporates CSA and GSA. Specifically, in Figure 8, the first and third rows exhibit cases of false detections, which arise from the lack of effective feature extraction operations. In the second row, illustrations (a) and (b) show cases of missed detections, attributed to dense object interference that hampers feature distinction between foreground and background or feature overlap. It is worth noting that in illustrations (c) and (d) of the second row, a small target in the bottom-right corner is detected, even though it is not annotated in the ground truth (GT), which demonstrates that the model incorporating global attention can achieve better detection performance for small targets.

*4.4. Benchmark Comparisons*

We conduct comparative experiments using three variants of MVT on the EOD, Vis-Drone, and UAVDT datasets, benchmarking against state-of-the-art methods with mAP.

4.4.1. Results on the EOD Dataset

We compare our method with several state-of-the-art detectors as shown in Table 5. MVT-B achieves 28.7% mAP@0.5:0.95, 56.5% mAP@0.5, and 26.3% mAP@0.75 on the EOD dataset, outperforming all other state-of-the-art methods. Figure 9 presents the results of our method for detecting objects in various scenes within the EOD dataset. As we expected, MVT has demonstrated superior detection performance for small objects, achieving 16.6% $mAP_S$.

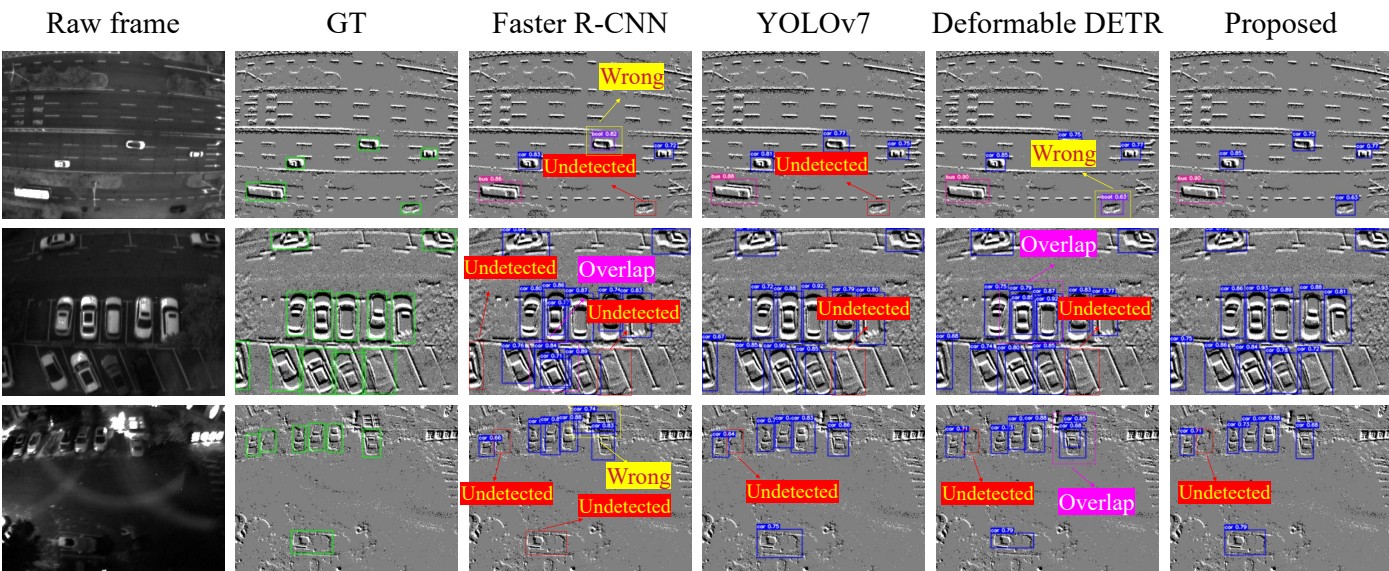

**Figure 9.** Prediction examples on the EOD dataset using different approaches involving Faster R-CNN, YOLOv7, Deformable DETR, and proposed method.

**Table 5.** Comparison of detection performance on the EOD dataset. The best result is highlighted with **bold**.

| Model | Backbone | mAP @0.5:0.95 | mAP @0.5 | mAP @0.75 | $mAP_S$ | $mAP_M$ | $mAP_L$ | Params |
|---|---|---|---|---|---|---|---|---|
| Faster R-CNN [34] | ResNet 50 | 0.183 | 0.392 | 0.122 | 0.089 | 0.202 | 0.371 | 42.0 M |
| DetectoRS [40] | ResNet 101 | 0.194 | 0.433 | 0.154 | 0.103 | 0.235 | 0.389 | 540.1 M |
| YOLOv5 [27] | CSPDarkNet 53 | 0.232 | 0.469 | 0.190 | 0.113 | 0.263 | 0.466 | 93.0 M |
| Cascade R-CNN [28] | Transformer | 0.234 | 0.445 | 0.208 | 0.122 | 0.276 | 0.485 | 335.0 M |
| YOLOv7 [41] | CSPDarkNet 53 | 0.237 | 0.480 | 0.197 | 0.118 | 0.286 | 0.479 | 135.8 M |
| DMNet [42] | CSPDarkNet 53 | 0.255 | 0.503 | 0.228 | 0.142 | 0.311 | 0.529 | 96.7 M |
| Sparse R-CNN [43] | Transformer | 0.259 | 0.510 | 0.215 | 0.133 | 0.312 | 0.521 | 352.0 M |
| Deformable DETR [3] | ResNet 50 | 0.262 | 0.521 | 0.238 | 0.145 | 0.317 | 0.536 | 41.0 M |
| CLusDet [44] | ResNeXt 101 | 0.266 | 0.543 | 0.244 | 0.127 | 0.332 | 0.547 | - |
| MVT-B (ours) | Transformer | **0.287** | **0.565** | **0.263** | **0.166** | **0.353** | **0.580** | 106.4 M |
| MVT-S (ours) | Transformer | 0.273 | 0.557 | 0.255 | 0.159 | 0.341 | 0.551 | 56.6 M |
| MVT-T (ours) | Transformer | 0.258 | 0.525 | 0.230 | 0.144 | 0.315 | 0.542 | **26.1** M |

4.4.2. Results on VisDrone2019 Dataset

We compare our method with several state-of-the-art detectors as shown in Table 6. MVT-B achieves 31.7% mAP@0.5:0.95, 52.2% mAP@0.5, and 34.2% mAP@0.75 on the VisDrone2019 dataset, outperforming all other state-of-the-art methods expect mAP@0.5.

Figure 10 presents the results of our method for detecting objects in various scenes within the VisDrone2019 dataset.

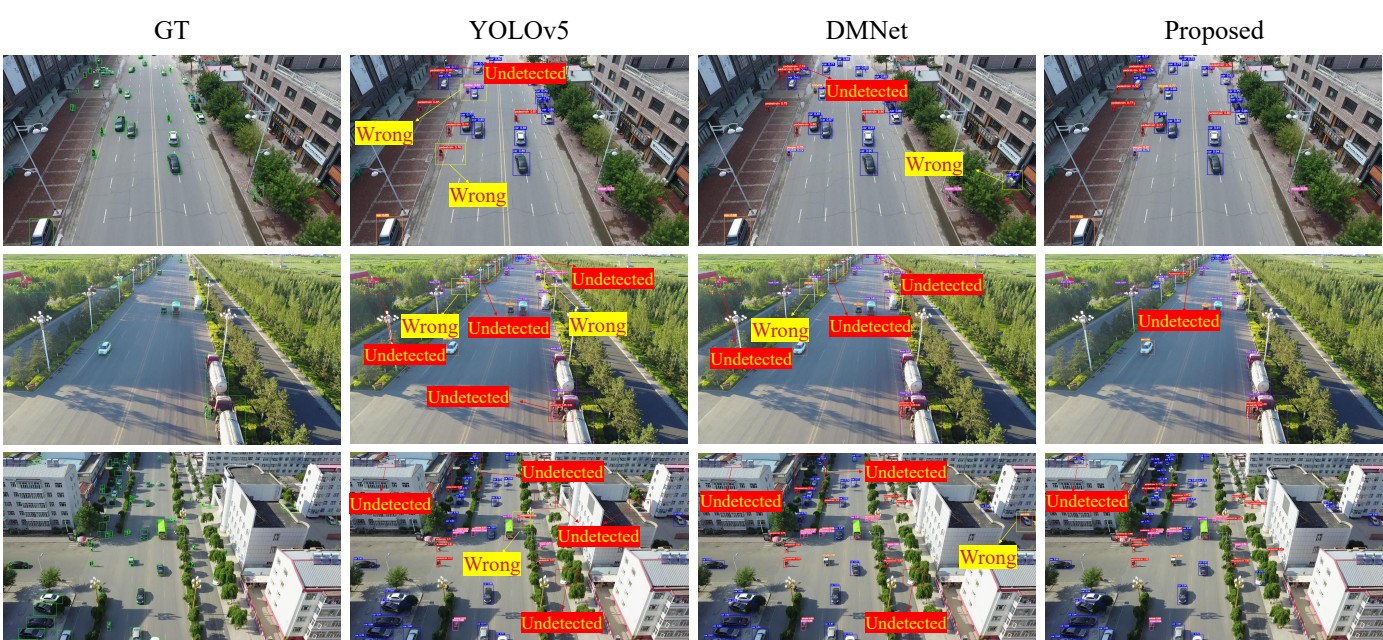

**Figure 10.** Prediction examples on the VisDrone2019 dataset using different approaches involving YOLOv5, DMNet, and proposed method.

**Table 6.** Comparison of detection performance on the VisDrone2019 dataset. The best result is highlighted with **bold**.

| Model | Backbone | mAP @0.5:0.95 | mAP @0.5 | mAP @0.75 | $mAP_S$ | $mAP_M$ | $mAP_L$ | Params |
|---|---|---|---|---|---|---|---|---|
| Cascade R-CNN [28] | ResNet 50 | 0.232 | 0.399 | 0.234 | 0.165 | 0.368 | 0.394 | 273.2 M |
| YOLOv5 [27] | CSPDarknet 53 | 0.241 | 0.441 | 0.247 | 0.153 | 0.356 | 0.384 | 93.0 M |
| RetinaNet [45] | ResNet 101 | 0.243 | 0.443 | 0.187 | 0.187 | 0.352 | 0.378 | 251.7 M |
| Libra RCNN [29] | ResNet 50 | 0.243 | 0.412 | 0.249 | 0.168 | 0.340 | 0.368 | 185.4 M |
| Cascade R-CNN [28] | Transformer | 0.247 | 0.424 | 0.265 | 0.177 | 0.372 | 0.403 | 335.0 M |
| HawkNet [46] | ResNet 50 | 0.256 | 0.443 | 0.258 | 0.199 | 0.360 | 0.391 | 130.9 M |
| VFNet [47] | ResNet 50 | 0.259 | 0.421 | 0.270 | 0.168 | 0.373 | 0.414 | 296.2 M |
| DetectoRS [40] | ResNet 101 | 0.268 | 0.432 | 0.280 | 0.175 | 0.382 | 0.417 | 540.1 M |
| Sparse R-CNN [43] | Transformer | 0.276 | 0.463 | 0.282 | 0.188 | 0.392 | 0.433 | 352.0 M |
| DMNet [42] | CSPDarknet 53 | 0.282 | 0.476 | 0.289 | 0.199 | 0.396 | **0.558** | 96.7 M |
| ClusDet [44] | ResNeXt 101 | 0.284 | **0.532** | 0.264 | 0.191 | 0.408 | 0.544 | - |
| SDMNet [48] | CSPDarknet 53 | 0.302 | 0.525 | 0.306 | 0.226 | 0.396 | 0.398 | 96.6 M |
| MVT-B (ours) | Transformer | **0.317** | 0.522 | **0.342** | **0.243** | **0.421** | 0.552 | 106.4 M |
| MVT-S (ours) | Transformer | 0.296 | 0.497 | 0.321 | 0.225 | 0.405 | 0.533 | 56.6 M |
| MVT-T (ours) | Transformer | 0.277 | 0.465 | 0.303 | 0.202 | 0.388 | 0.502 | **26.1** M |

### 4.4.3. Results on UAVDT Dataset

We compare our method with several state-of-the-art detectors as shown in Table 7. MVT-B achieves 28.2% mAP@0.5:0.95, 42.1% mAP@0.5, and 32.2% mAP@0.75 on the UAVDT dataset, outperforming all other state-of-the-art methods expect mAP@0.5. Figure 11 presents the results of our method for detecting objects in various scenes within the UAVDT dataset.

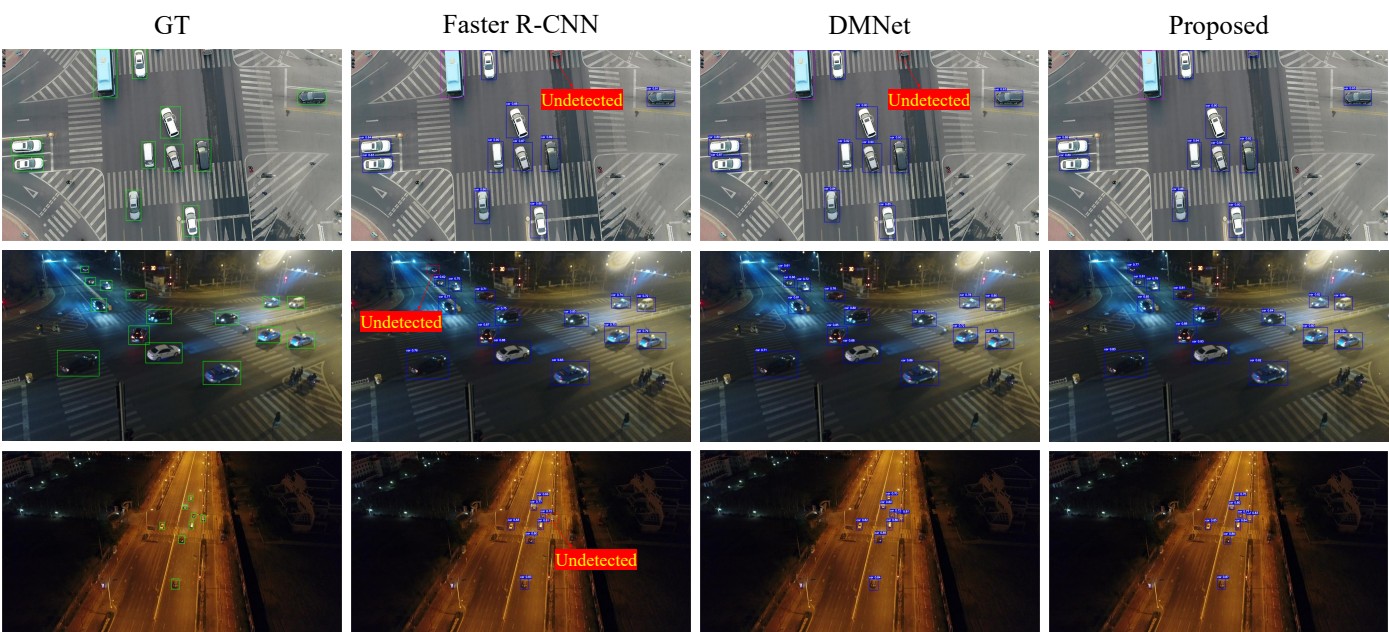

**Figure 11.** Prediction examples on the UAVDT dataset using different approaches involving Faster R-CNN, DMNet, and proposed method.

**Table 7.** Comparison of detection performance on the UAVDT dataset. The best result is highlighted with **bold**.

| Model | Backbone | mAP @0.5:0.95 | mAP @0.5 | mAP @0.75 | mAP$_S$ | mAP$_M$ | mAP$_L$ | Params |
|---|---|---|---|---|---|---|---|---|
| Faster R-CNN [34] | ResNet 50 | 0.110 | 0.234 | 0.084 | 0.081 | 0.202 | 0.265 | 42.0 M |
| Cascade R-CNN [28] | ResNet 50 | 0.121 | 0.235 | 0.108 | 0.084 | 0.215 | 0.147 | 273.2 M |
| ClusDet [44] | ResNet 101 | 0.137 | 0.265 | 0.125 | 0.091 | 0.251 | 0.312 | - |
| Cascade R-CNN [28] | Transformer | 0.138 | 0.244 | 0.117 | 0.090 | 0.232 | 0.268 | 335.0 M |
| DMNet [42] | CSPDarkNet 53 | 0.147 | 0.246 | 0.163 | 0.093 | 0.262 | 0.352 | 96.7 M |
| Sparse R-CNN [43] | Transformer | 0.153 | 0.266 | 0.171 | 0.118 | 0.253 | 0.288 | 352.0 M |
| GLSAN [49] | CSPDarkNet 53 | 0.170 | 0.281 | 0.188 | - | - | - | - |
| AdaZoom [50] | CSPDarkNet 53 | 0.201 | 0.345 | 0.215 | 0.142 | 0.292 | 0.284 | - |
| ReasDet [51] | CSPDarkNet 53 | 0.218 | 0.349 | 0.248 | 0.153 | 0.327 | 0.308 | - |
| EVORL [52] | ResNet 50 | 0.280 | **0.438** | 0.315 | 0.218 | **0.404** | 0.359 | - |
| MVT-B (ours) | Transformer | **0.282** | 0.421 | **0.322** | **0.237** | 0.397 | **0.368** | 106.4 M |
| MVT-S (ours) | Transformer | 0.267 | 0.405 | 0.297 | 0.206 | 0.373 | 0.350 | 56.6M |
| MVT-T (ours) | Transformer | 0.238 | 0.367 | 0.271 | 0.162 | 0.356 | 0.322 | **26.1M** |

## 5. Discussion

The UAVDT dataset only annotates three categories of objects and has simpler scenes compared to the VisDrone2019 dataset. However, the UAVDT dataset exhibits lower detection performance due to its challenging scenes (e.g., low lighting, motion blur), as exemplified in Figure 12. Therefore, applying event cameras to improve visual effects in extreme environments will greatly improve the accuracy of object detection. Despite event cameras being capable of capturing moving objects in various challenging scenarios, they only retain intensity features while losing color information, resulting in the loss of object details. While traditional cameras are limited by a fixed frame rate, they preserve more high-frequency information. Therefore, it is a meaningful step to simultaneously consider the event and traditional cameras for detection, aiming to achieve improved performance in any challenging scenario.

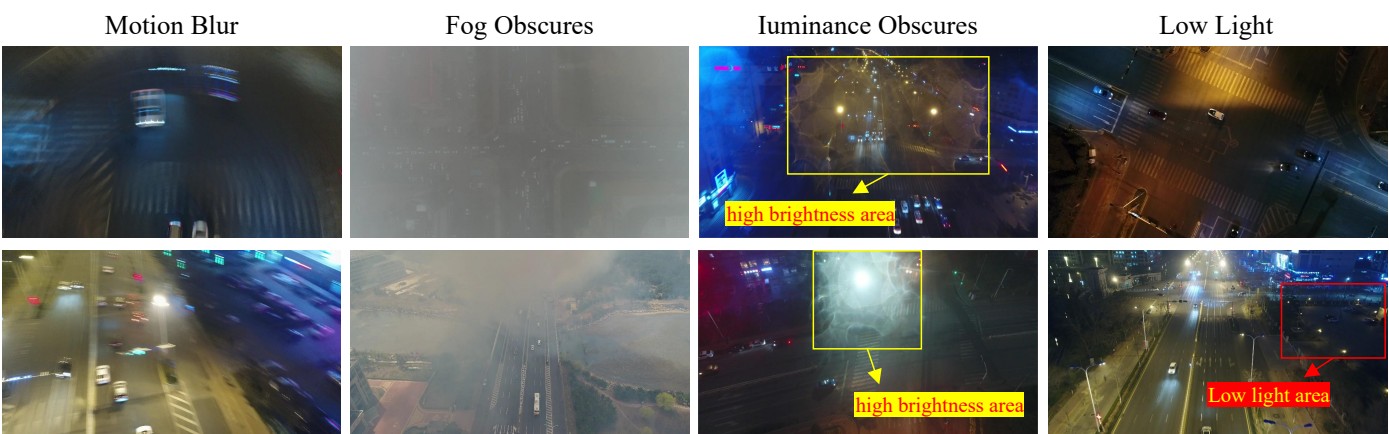

**Figure 12.** Extreme scenarios in UAVDT dataset. These scenes captured by traditional cameras pose challenges for object detection.

## 6. Conclusions

In this paper, we aim to capture details in challenging remote sensing images (e.g., low light, motion blur scenarios) to improve the detection performance of small targets. We propose a method called Multi-Vision Transformer (MVT), which employs Channel Spatial Attention (CSA) to enhance short-range dependencies and extract high-frequency information features, utilizing Global Spatial Attention (GSA) to strengthen long-range dependencies and retain more low-frequency information. Specifically, the proposed MVT backbone generates more accurate object locations with enhanced features by maintaining multi-scale high-resolution features with rich semantic information. Subsequently, we use Scale-Level Embedding to extract multiple scales features and apply Cross Deformable Attention (CDA) to progressively fuse information from different scales, significantly reducing the computational complexity of the network. Furthermore, we introduce a dataset called EOD, captured by a drone equipped with an event camera. Finally, all experiments are conducted on the EOD dataset and two widely used UAV remote sensing datasets. The results demonstrate that our method outperforms widely used methods in terms of detection performance on the EOD dataset, VisDrone2019 dataset, and UAVDT dataset.

**Author Contributions:** Conceptualization, methodology, software, S.J.; data curation, visualization, investigation, H.L. (Hengyi Lv); software, validation, Y.Z.; software, Writing—original draft preparation, H.L. (Hailong Liu); writing—reviewing and editing, M.S. All authors have read and agreed to the published version of the manuscript.

**Funding:** This work was supported by the National Natural Science Foundation of China (62005269) and the 2023 Jilin Province industrialization project for the specialized program under Grant 2023C031-6.

**Data Availability Statement:** The VisDrone2019 dataset and UAVDT dataset are available from the websites https://github.com/VisDrone/VisDrone-Dataset and https://sites.google.com/view/grli-uavdt.

**Acknowledgments:** The authors thank the editors and reviewers for their hard work and valuable advice.

**Conflicts of Interest:** The authors declare no conflicts of interest.

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
