# Peer review of "MVT: Multi-Vision Transformer for Event-Based Small Target Detection"

_remotesensing, doi:10.3390/rs16091641_

Round 1
Reviewer 1 Report
Comments and Suggestions for Authors
This paper proposes a method called Multi Vision Transformer (MVT), which employs Channel Spatial Attention (CSA) to enhance short-range dependencies and extract high-frequency information features, utilizing Global Spatial Attention (GSA) to strengthen long-range dependencies and retain more low-frequency information. Despite some contributions, there are many problems exist. These issues are listed below.
1. The author needs to carefully check the paper. During reading, I found some errors in word capitalization, sentence tenses, and punctuation.
2. The author can conduct more in-depth research and analysis on remote sensing image object detection methods. When using event cameras for remote sensing image object detection, what are the shortcomings of existing methods and the reasons for their poor performance? Section 2.3 does not provide much analysis on related work. Has the author conducted a comprehensive research on other event camera based object detection algorithms? The author can add descriptions to demonstrate the superiority of the method proposed in this article.
3. The backbone used in the paper is VIT, but all the backbones used in the comparative experiments are resnet. It is recommended to add comparative experiments to compare with other VIT based object detection methods.
Comments on the Quality of English LanguageThe English writing of the paper is relatively good, but there are still some details that need to be carefully checked and revised by the author.
Reviewer 2 Report
Comments and Suggestions for Authors
This manuscript proposes a remote sensing detection network MVT and an event camera-based dataset EOD with full ablation experiments, which is novel and interesting. Overall, the manuscript is well written and the results are convincing.
However, I have some of the following questions:
1. CDA seems to be very effective, and the author provides an attention visualization comparison. I think adding the detection results in Figure 7 will be more convincing.
2. The author mentioned that CDA can reduce the computational complexity of Transformer Encoder by about 82%, but Table 2 only has overall GFLOPs. Please add the encoder GFLOPs.
3. The writing of the article is good, but this part of the introduction needs polishing to make it easier to understand: event cameras independently measure and output the logarithmic intensity changes of each pixel, also, the main distinction lies in the continuous output of asynchronous event streams by event cameras, as opposed to capturing images. When it comes to capturing fast-moving objects, traditional cameras require a significant cost to achieve satisfactory results. In contrast, event cameras can effectively circumvent the limitations imposed by fixed frame rates, providing asynchronous information with sub-millisecond latency, thereby offering superior performance. As a result, event cameras possess characteristics such as low latency, low power consumption, high dynamic range, and high temporal resolution. In addition, due to the fact that event cameras only capture changes in light intensity at different positions, they possess an extremely high dynamic range, allowing them to capture objects even in low-light conditions or extremely bright lighting.
4. The contribution part in the Introduction should be streamlined.
5. The regression block in Figure 2 is easy to cause misunderstanding, and the author needs to modify the classification part of the detection head.
6. Does Formula 11 require softmax first and then matrix multiplication? Is the writing method wrong? Please explain.
7. The author needs to unify the units of Params in the Table. For example, the units of params in Table 2 should be consistent with the writing method of other Tables.
8. The author should add explanations of evaluation indicators such as mAP@0.5:0.95, mAP@0.5, mAPS, mAPM, mAPL.
Reviewer 3 Report
Comments and Suggestions for Authors
Object detection in aerial images (drones). Vision transformers designed: MVT and CDA.
Event camera (new camera with dynamic sensor). It captures pixel noticeable differences.
Own dataset EOD (Event Object Detection), use of known datasets: VisDrone2019 and UAVDT Dataset. Six categories of objects: "e car, truck, pedestrian, two-wheel, boat, and ship". Is EOD dataset public?
DETAILS:
========
Please clarifiy:
- The sign in equation 1 is "equal" (=) or "greater than or equal" (>=)?
- What is Et(N) in equation 2? N?
- What is layer norm in figure 4? Is it possible to put tensors sizes also in figure 4?
- The division of datasets into training/validation/testing is the same for all tests?
Comments on the Quality of English Language
Overal revision needed to correct minor errors, examples:
Line 23 seems non-coherent, final "working in a new paradigm" seems uncoonected with the previous.
In line 79: "Then, we employing MVT-B trained for 36 epochs, achieving 31.7% mAP@0.5:0.95 and 24.3% APS on the VisDrone2019 Dataset, 28.2% mAP@0.5:0.95 and 23.7% APS on the UAVDT Dataset".
I would write: "in this case, we employ...".
Line 123: "As a result, there are various improvement methods have been developed."
I would write: "For this reason various improvements..."
Line 206: "There are several studies have shown that enhancing multi-scale features can 206 significantly improve small target detection."
I would say: "several studies have shown...".
Line 245: "afterwards"... with "s".
